# Safety and Efficacy of Wharton’s Jelly Connective Tissue Allograft for Rotator Cuff Tears: Findings from a Retrospective Observational Study

**DOI:** 10.3390/biomedicines12040710

**Published:** 2024-03-22

**Authors:** Albert Lai, Conrad Tamea, John Shou, Anthony Okafor, Jay Sparks, Renee Dodd, Crislyn Woods, Naomi Lambert, Orion Schulte, Tyler Barrett

**Affiliations:** 1Desert Physical Medicine and Pain Management, Indio, CA 92201, USA; albertlaimd@gmail.com (A.L.); renee@prestigepro.net (R.D.); 2Orthopedic Associates of Tampa Bay, Tampa, FL 33603, USA; conradtameamd@gmail.com; 3Pharmacology, Baylor College of Medicine, Houston, TX 77030, USA; 4Mathematics & Statistics, University of West Florida, Pensacola, FL 32514, USA; aokafor@uwd.edu (A.O.); jsparks@uwf.edu (J.S.); 5Regenative Labs, Pensacola, FL 32501, USA; crislyn@regenativelabs.com (C.W.); orion@regenativelabs.com (O.S.); tyler@regenativelabs.com (T.B.)

**Keywords:** rotator cuff, Wharton’s Jelly, regenerative medicine, umbilical cord tissue

## Abstract

With the increasing occurrence of rotator cuff injuries every year, there is a great need for a reliable treatment option. Wharton’s Jelly contains several components that can positively impact the replacement and repair of musculoskeletal defects. The overall objective of this study is to evaluate the improvement of patient-reported pain scales after applying Wharton’s Jelly (WJ) in rotator cuff defects. Eighty-seven patients with rotator cuff defects who failed at least eight weeks of conservative treatment were selected from the retrospective repository. A total of 2 cc of WJ flowable allograft was applied to the specific affected anatomy, the most common being supraspinatus tendon, biceps tendon insertion, labral tear, and subscapularis tear. No adverse reactions were reported. Statistically significant improvements were found from the initial to Day 90 in all scales. Patient satisfaction was calculated using minimal clinically important differences. No statistically significant differences were found in mean changes between gender, BMI, and age. Scanning electron microscopy images reveal the similarities between the collagen matrix in WJ and the rotator cuff. The significant improvement in patient outcomes coincides with the current literature analyzing WJ applications with other structural defects around the body. WJ is a promising alternative for musculoskeletal defects when the standard of care fails.

## 1. Introduction

There are approximately 4.5 million shoulder pain patient visits and 250,000 rotator cuff (RC) repairs annually at continuously increasing rates in the United States [1]. The RC provides a wide range of shoulder movement and stabilizes the glenohumeral joint through the contraction of the subscapularis, infraspinatus, teres minor, and supraspinatus muscles [2]. Shoulder pain and weakness are often associated with shoulder conditions, including RC disorders, adhesive capsulitis, superior labrum, bicep lesions, acromioclavicular joint disease, and instability [3]. The shoulder is the most mobile and unstable joint in the body as it is a ball-and-socket joint that forms an extremely shallow articulation, making it susceptible to injury [3]. RC injuries can be diagnosed through physical examination and imaging modalities. Diagnostic imaging often includes radiography, ultrasound, and MRI. Common causes of RC tears include falls, shoulder dislocation, violent pull or sudden traction injury, direct trauma or impact to the shoulder, hyperextension, and lifting heavy objects. Factors that increase the risks of RC tears include age, hand dominance, history of trauma, nicotine use, hypercholesterolemia, and genetics [1]. Regarding the impact of age on RC defects, they have been found to affect up to 70% of people over 70 [4]. Elderly patients often present multiple risk factors, including decreased bone quality, poor blood supply, and increased medical comorbidities [5]. Diabetes and other systemic disorders have also been associated with a greater risk of RC defects [6]. The structural integrity of the connective tissue matrix found in the RC is imperative when considering repair.

Current treatment for RC tears primarily exists as nonsurgical or surgical. Asymptomatic tears are typically managed through nonsurgical techniques determined by tear thickness, size, and morphology [7]. The most common nonsurgical treatment plans include NSAIDS, physical therapy, and injections. The most common injections include platelet-rich plasma (PRP) and hyaluronic acid (HA) [8,9]. These modalities can also be accompanied by extracorporeal shockwave therapy (ESWT) to increase hypervascularity [10]. A study by Maman et al. (2009) found that patients older than 60 have a 54% deteriorating rate compared to younger patients with a 17% deteriorating rate [11]. Although conservative treatments have a typically high rate of success, the older population takes much longer to recover, with high risks of re-injury. When non-surgical treatments are unsuccessful, surgical operations are typically undergone to resolve RC tears. Some major concerns among orthopedic surgeons from rotator cuff surgery include high re-tear rates and decreased range of motion [12]. The rate of re-tears has shown a strong association between the strength of the repair, the tear size, and the tissue quality of the tendon [12]. Re-tears are common as the native tissue contains type I collagen fibers. In contrast, the repaired muscle contains more type III collagen fibers, which are more disorganized and have reduced tensile strength. Studies have been conducted to indicate that, as the size of the tear increases, the likelihood of regaining the full range of motion of the affected shoulder decreases. Likewise, it is expected to take one year to regain external rotation after small and medium tears [13]. Given the time it takes for repair and the cost of treatment, the total cost of rotator cuff repair must be considered. Annually, there are more than 250,000 rotator cuff repairs in the United States, accounting for an estimated USD 1.2 to 1.6 billion in healthcare expenditures [14]. As there is no conclusive evidence for the best treatment and given the significant economic burden rotator cuff repairs have on the healthcare system, it is clear there is a need for further treatment research regarding RC repairs.

Wharton’s Jelly (WJ) primarily comprises collagen fiber types I, III, and V, cytokines, proteoglycans, various growth factors, and hyaluronic acid. In general, each component of the WJ composition can positively impact the repair of musculoskeletal injuries [15]. The collagen fibers present in WJ are comparable to the extracellular matrices of human articular cartilage, tendons, and dermal tissues [16]. Studies suggest that WJ is also efficacious in other areas of the body. WJ has shown promising results in the application to defects of the sacroiliac (SI) joint. When WJ was applied to the SI joint, there were statistically significant improvements in function, joint mobility, and pain relief [17]. In addition, when WJ was applied to the knee, data were analyzed to show pain alleviation, function improvement, and a potential delay in total knee replacement in patients with knee osteoarthritis [18]. While WJ has been successfully applied in humans in over 180 different homologous use sites, research has yet to be published on its application on defects of the human rotator cuff. However, a study by Yuan (2022) utilized WJ in a rabbit rotator cuff tendon defect and produced statistically significant data on improved tendon healing and enhanced biomechanical strength of repaired tendons [19]. There is a severe need for alternative medical intervention with a lack of core evidence for long-term RC improvement through standard interventions. This study aims to observe the efficacy and safety of Wharton’s Jelly tissue allografts applied to structural defects of the rotator cuff.

## 2. Materials and Methods

### 2.1. Study Design

This study cohort is derived from the observational retrospective repository at Regenative Labs. The data repository is in accordance with the Declaration of Helsinki and has maintained approval from the Institutional Review Board of the Institute of Regenerative and Cellular Medicine (IRCM-2022-311) since May 2021. The repository protocol includes observational data collection of patients who provide informed consent and receive one or more applications of either their Wharton’s Jelly tissue allografts ProText^TM^, CryoText^TM^, SecreText^TM^, or their dehydrated amniotic membrane allograft, AmnioText^TM^. All patients in the repository have documented failure of standard of care practices for at least 90 days regardless of use site, making them candidates for Wharton’s jelly tissue allografts. All methods of tissue processing at Regenative Labs, Pensacola, FL, USA, comply with the FDA and American Association of Tissue Banks (AATB) standards. Greater details of tissue allograft production can be found on the manufacturer’s website. The study design for this paper identified the unique patients from the repository who had one application to either the tendons or muscles of the rotator cuff (infraspinatus, supraspinatus, subscapularis, teres minor). The inclusion and exclusion criteria can be found in the study flowchart below (Figure 1).

### 2.2. Study Population

The patient datasets for this study were collected from the retrospective repository from the research department at Regenative Labs. The inclusion criteria are as follows: patients with rotator cuff-related defects, at least one 150 mg WJ tissue application, and a full 0, 30, and 90-day dataset completed within the allotted time constraints. Exclusions were based on incomplete or incorrect data submissions, not on age, gender, or BMI. There was a total of 87 patients in this study. The population was made up of 42 females and 45 males. The average age of the population was 71 years old, with the youngest patient being 36 years old and the oldest patient 89 years old. The average BMI was 27.7, with the smallest being 17.1 and the greatest 38.6. Table 1 shows the demographic characteristics of participants.

### 2.3. Allograft Application

Twenty private practices obtained ProText^TM^, an umbilical cord tissue allograft. When purchased, the WJ allograft arrives frozen on dry ice in 2 mL vials and must be stored at −40 °C or colder until ready to apply. Each vial of ProText^TM^ contains 150 mg of WJ tissue minimally manipulated into 300-micron particles suspended in sterile saline with 5% dimethyl sulfoxide to act as a cryoprotectant. Patients underwent physical examination, and their medical history was evaluated to ensure at least eight weeks of failed conservative management. Each patient had evidence of structural degeneration of the symptomatic rotator cuff confirmed by either ultrasound or MRI. All patients failed conservative treatments that may have included NSAIDs, muscle relaxants, physical therapy, pain medications, and steroid or PRP injections. Symptoms at the site of application were confirmed, and informed consent was obtained on the day of the WJ application. Twenty minutes before the application, the tissue was removed from the freezer or dried ice packaging to defrost. The vial was then inverted several times to mix and ensure complete suspension of the tissue particles. Under sterile technique and ultrasound guidance, 2 cc of Wharton’s jelly flowable allograft, or 150 mg of Wharton’s Jelly, was applied to the specific affected anatomy, the most common being supraspinatus and subscapularis tendon. Patients were monitored for 30 min post-procedure. No patients experienced post-procedure complications and were all discharged home in stable condition with instructions to maintain an active and passive range of motion of the shoulder and to avoid strenuous activity.

### 2.4. Questionnaire Composition

Patients filled out a questionnaire on the day of application, consisting of the numerical pain rating scale (NPRS), the Western Ontario and McMaster Universities arthritis index (WOMAC), and quality of life scales (QOLS). As this sample is from a large retrospective repository that initially focused on lower extremities, the WOMAC is not ideal for evaluating shoulder-related issues. However, it is still relevant because the patient reports their pain evaluation based on the defect relevant to their WJ application. Patients answered the same questionnaire 30 and 90 days after the initial allograft application. The scores of these two scales were analyzed individually to allow for a more significant examination of the physical mobility of the affected joint. 

### 2.5. Statistical Analysis

Repeated measures analysis of variance (ANOVA) was performed to assess the difference within groups in outcomes between intervals. This is because the pain scales recorded three different time intervals as continuous measures, and the values were normally distributed. Depending on the ANOVA results, Tukey’s test was employed to define which groups differed from each other. Logistics regression was performed for the binary (or classified) outcomes. All data analyses used the programming language R version 4.2.

## 3. Results

Results were obtained from three intervals, including the initial application day, day 30, and day 90. The six scales utilized to quantize pain were the NPRS, QOLS, WOMAC, and the three WOMAC subsections of pain, stiffness, and function. Shown in the table below (Table 2, Figure 2) are the calculated mean and standard deviation (SD) of the six pain scales for each interval. Higher values in NPRS, WOMAC, pain, stiffness, and functionality indicate more significant pain, whereas higher values in QOLS indicate a better quality of life (not necessarily related to pain). The mean scores of the six pain scales declined over time from the Initial up to 30 days and 90 days, with the most significant decline appearing between the Initial and Day 90 after the WJ application. The improvement percentage in the patient-reported scales displayed in Figure 3 reflects the decrease in numerical scores, indicating an improvement in pain levels.

Repeated measures analysis of variance (ANOVA) and Tukey’s test were employed for continuous values, and logistic regression was performed for binary (or classified) outcomes. ANOVA tested the significance level of changes between intervals for the six pain scales. After determining these, test results suggested there were differences between intervals for WOMAC, pain, stiffness, and functionality. Accordingly, Tukey’s test was performed to identify the specific intervals that differed from each other. Table 3 shows the actual differences in the six scales between each interval with 95% CI and *p*-values.

There were statistically significant differences between Initial and Day 90 for all six scales. The difference between Initial and Day 30 was significant only for NPRS and stiffness. A *p*-value of (0.05) was used to assess the degree of significance.

ANOVA tested the significance level of the effect of changes for any demographic variables on the six scales. However, there was no effect on the six scales based on gender, BMI, and age. Entries in Table 4 below are the *p*-values for six scales.

The anchor-based method with receiver operating characteristic (ROC) analysis was used to summarize the minimal clinically important difference (MCID). The NPRS score was recorded at the beginning of the study and rerecorded 90 days after their first injection to define anchor groups as a baseline. After removing the missing NPRS scores for Initial and Day 90, a total of 66 patients for the four pain scales and 59 for QOLS were used to determine the anchor groups. The difference in NPRS score between Initial and Day 90 was used to calculate the difference over time. The anchor question can be expressed as “How different is your pain compared to before and after the injection?” After determining the changes, the patients were grouped into four categories by the range of changes in their NPRS scores. The grouped answers were “Not better”, negative-zero, “Slightly better”, 1–3, “Better”, 4–5, and “Much better” 6–9. Table 5 shows the descriptive statistics between the answers.

The ROC analysis determined the probability of being “Not better” or “Slightly better” and calculated the best cutoff and area under the curve (AUC) based on the highest Youden’s index with the best sensitivity and specificity. The mean changes for “Slightly better” in the MCID for the five scales were estimated by choosing Youden’s index which gave the highest AUC values (see Table 6).

The AUC values were 0.88 for WOMAC (see Figure 4), 0.79 for pain, 0.79 for stiffness, 0.83 for functionality, and 0.61 for QOLS. Based on the reliability of AUC, the MCID for QOLS is not meaningful in this study.

The meaningful MCIDs were 10.0 for WOMAC, 2.36 for pain, 1.21 for stiffness, and 6.43 for functionality for the anchor group (see Table 7).

## 4. Discussion

The patients in this retrospective study achieved successful results in 90 days after failing standard-of-care practices for at least eight weeks. Typically, patients who fail nonoperative treatment do so in the first 12 weeks [20]. For all age ranges, most studies show a success rate of 75% for nonoperative treatment [21]. The sample patient population used in this study represents patients in the remaining 25% who do not benefit from physical therapy and saw minimal improvement in the standard care amount of time. Instead of proceeding with a surgical procedure after failed nonoperative treatments, the patients in this study cut costs and recovery time by not having surgical intervention and subsequent rehab or medications. Another significant factor is that the mean age of the sample is 71 years old, and the statistics showed no differentiation in improvement relating to age. When a patient reaches 60 years or older, the current literature shows the progression of a rotator cuff tear or retear increases significantly, further complicating treatment and diminishing success rates.

Although all patients who received an application had failed the standard of care for eight weeks minimum, one limitation of this study is the lack of a control group to compare improvements to due to the nature of the retrospective repository used. Looking at the current literature, we can compare similar patient groups with other conservative treatments and placebo groups. A study by Annaniemi in 2022 compares platelet-rich plasma (PRP) and corticosteroid (CS) injections for rotator cuff tendinopathy. The patients either received three PRP injections or one CS and were tracked at intervals of 6, 12, and 18 months [22]. There was a total of 75 patients aged 18–90. Both groups showed improvement in the Western Ontario Rotator Cuff Index (WORC), Visual Analog Scale (VAS), and Range of Motion (ROM), but there was no significant difference between the groups. Their study did not have a 3-month follow-up point, but the median improvement percentage between the initial visit and their 6-month follow for the PRP group and CS group was approximately 27% and 32%, respectively, for WORC, and 30% and 34%, respectively, for VAS. A systematic review and meta-analysis by Bansal in 2023 compared hyaluronic acid (HA) injections to PRP, CS, and placebo groups from 18 randomized control trials (RCTs) [9]. They found that there was significant improvement (*p* < 0.05) in VAS and constant Murphy score when comparing HA with physical therapy and PRP on short-term follow-up (1 and 3 months), with insignificant results on comparing HA with placebo (normal saline) and steroids. When comparing to our results from one application of WJ, the NPRS that is comparable to VAS may have improved better than other noninvasive modalities while the WOMAC, comparable to the WORC, had similar results. Although pain is relative, one study found that long-term radiographic assessment after rotator cuff repair reveals direct correlations between failure and patient-reported outcomes, functional deterioration, progression of arthritis, and/or frank cuff tear arthropathy [23]. As previously mentioned, the WOMAC is not the ideal scale for rotator cuff defects, so further studies with the commonly used WORC can be performed to better understand the efficacy of WJ compared to other modalities like PRP, CS, HA, and placebo groups.

The direct and indirect healthcare cost of rotator cuff repair postoperatively accumulates to approximately USD 438,892,670 for a short-term period [24]. The average cost for a patient to receive revisional rotator cuff surgery is USD 17,098 per patient. Additionally, the risk of postoperative infection and stiffness exists. The estimated national healthcare cost for postoperative complications is USD 2,504,873. A study completed by Jangoo Kim analyzing nine different studies found that 11% to 94% of patients experience retear or healing failure after rotator cuff repair [25]. In comparison, the short-term total national healthcare cost for nonoperative management of failed rotator cuff repairs in 2022 was estimated to be USD 229,390,898, with an estimation of USD 2045 per patient. Based on a computational model that was developed in the study by Young, for every 5% improvement in the rate of successful structural healing of rotator cuff repairs in the United States, healthcare costs would decrease by more than USD 84 million [24]. This push toward structural repair is where the paradigm shift to regenerative medicine, targeting the root structural defect, can have a significant impact on healthcare costs.

One of the tests utilized to analyze the data was ANOVA. ANOVA tests whether the difference in mean changes between the groups is equal or not. Even though ANOVA results represent a significant difference in groups, they do not provide the exact pair of groups that are different. Therefore, Tukey’s test was used to identify which specific groups differed from each other. Tukey’s test was used as it functions to compare the mean of each treatment to the mean of all other treatments. In this study, ANOVA testes presented a difference between intervals for WOMAC and its subsections. Given this information, Tukey’s test was performed to determine how the intervals differed from each other. There was a statistically significant difference between the initial application and Day 90 for all six pain scales, as seen in Table 3, with *p* values less than 0.05 in bold. On average, each patient reported a 2.76 point reduction in NPRS score from the initial application to the 90 day follow-up, with raw scores ranging from 10-1 at the initial visit and 8-0 at the final visit. The highest possible total WOMAC score is 92 points; at the initial visit, total scores reported ranged from 92 to 4, and after the final 90-day visit, averaging a 13.5-point decrease, the reported sums ranged from 89 to 0. The QOLS has the highest possible score of 112, representing total satisfaction in all areas of life. The average improvement in QOLS scores increased by 8.9 points from the initial to final visits, with the initial range being 112-32 and ending at 112-41. It is important to note that, for the patient outcomes-focused clinic, 40% improvement is good but is not the goal; 100% is. Some of the patients in this cohort who did not achieve at least a 50% improvement may have opted to receive a second application under the guidance of their physician. While pain is subjective, these scales indicate significant improvement among the cohort. To best understand actual patient satisfaction, we included an MCID calculation to determine how meaningful this intervention was to each patient mathematically.

MCID is used to quantify the importance of the pain relief experienced by the patients [26,27,28]. The purpose of MCID is to identify the minimum and meaningful differences that are useful for the interpretation of a patient’s improvement. To establish the MCID on WOMAC (with subsections) and QOLS, the groups “Not better” and “slightly better” were used to define the MCID. These groups were not questions asked directly to patients but were determined statistically using the NPRS score as the anchor question. AUC was used to quantize the mean changes for “slightly better” in the MCID for the pain scales. Of all the scales, QOLS is the only scale that was not meaningful. AUC is considered to be meaningful if it has a value of 0.7 to 0.8 or above. Any value less than 0.7 is considered to be insignificant. This lack of change was expected as the quality of life scale used focused primarily on aspects of the patient’s life unrelated to physical pain. However, the WOMAC in its entirety and parts were found to be meaningful to the patients. Overall, 72% of the population of the study reported meaningful improvement by at least one unit, “slightly better”, and 51.7% exceeded the MCID_AUC,_ meaning they felt “better” or “much better.” Future studies may incorporate a quality-of-life scale that better evaluates how the patient’s injury affects the quality of life.

Throughout this study, no adverse reactions were reported, and the statistically significant improvements found in both NPRS and WOMAC scales show that WJ is safe and effective in its application to structural defects of the rotator cuff. These findings are consistent with the results of a knee osteoarthritis study in 2020, in which a total of 34 out of 42 patients reported significant clinical improvement [18]. A sacroiliac joint study in 2022 reported 32 of 38 patients with a lowered NPRS, and 29 of 38 patients reported a lowered WOMAC score [17]. In addition to the structural collagen matrix of WJ, the growth factors, cytokines, proteoglycans, and hyaluronic acid may be positive contributing factors to its success as a tissue transplant [16]. The perinatal tissue is considered immune privileged, not eliciting an immune response. The lack of immune response was reflected in the rotator cuff applications presented, with no adverse reactions reported from any patients. Combining the efficacy and safety of WJ observed, it is clear that WJ is an optimal option in the use of rotator cuff injuries.

Along with patient-reported improvement, the microstructures of the rotator cuff and WJ allografts can be observed for homologous structures. Collagen accounts for over 80% of the dry weight of human tendons, with the primary types being collagen I and III [29]. A study by Hashimoto et al. in 2003 performed a histopathologic examination revealing the thinned, snapped, and disorientated collagen fibers of the shoulder tendon in a 74-year-old patient (Figure 5) [30]. These collagen fibers were split longitudinally in primarily the deep layer of the tendon, with hyaline degeneration evident in the mid layer. The repair process of the tendon slows down with age, forming collagen type 3 first and then eventually repairing it with the sturdier collagen type 1. WJ, as pictured below, is primarily comprised of collagen type 1, then 3, then 5 (Figure 5 and Figure 6) [16], all vital to tendon repair; placing this tissue within the defect not only provides the body with the tissues that the body struggles to regenerate but also providing HA and other GAGs to promote this process. The current literature supports the success of using WJ in a homologous manner in knees, hips, and other areas to supplement damaged tissue, and that is further confirmed for rotator cuff uses in this study.

However, there are some limitations to this study. The WOMAC questionnaire used by participating clinics was designed for lower extremity use sites but still gives relevant insight into the patient’s pain during daily activities. When this study was initially designed, it was left open-ended to accept data on any homologous use site; the data form most clinics used was WOMAC. Future data collection in this retrospective repository will provide additional scales to account for the many homologous use sites discovered over the past three years. This study also covered all application sites within the rotator cuff that could be individually analyzed for more specific efficacy evaluations. Additionally, randomized studies will be beneficial for further confirming the positive results of Wharton’s jelly for rotator cuff defects and standardized application protocols.

## 5. Conclusions

The application of Wharton’s jelly allografts in this retrospective cohort was observed to promote decreases in patient-reported NPRS scores, pain, and stiffness in the shoulder and improve function. These improvements were consistent across age groups but were most notable for the elderly population, who generally have a slower and less effective recovery rate after conservative treatment and surgical procedures. Given the success shown by NPRS and WOMAC in decreasing pain in the elderly population, more studies are warranted to evaluate the safety and efficacy of Wharton’s Jelly, specifically in patients older than 65. The positive results from this study and the current literature provide a foundation to study applications of WJ in other musculoskeletal conditions to ultimately improve the quality of life and reduce the economic healthcare burden of multiple conservative treatments or expensive surgical procedures.

## Figures and Tables

**Figure 1 biomedicines-12-00710-f001:**
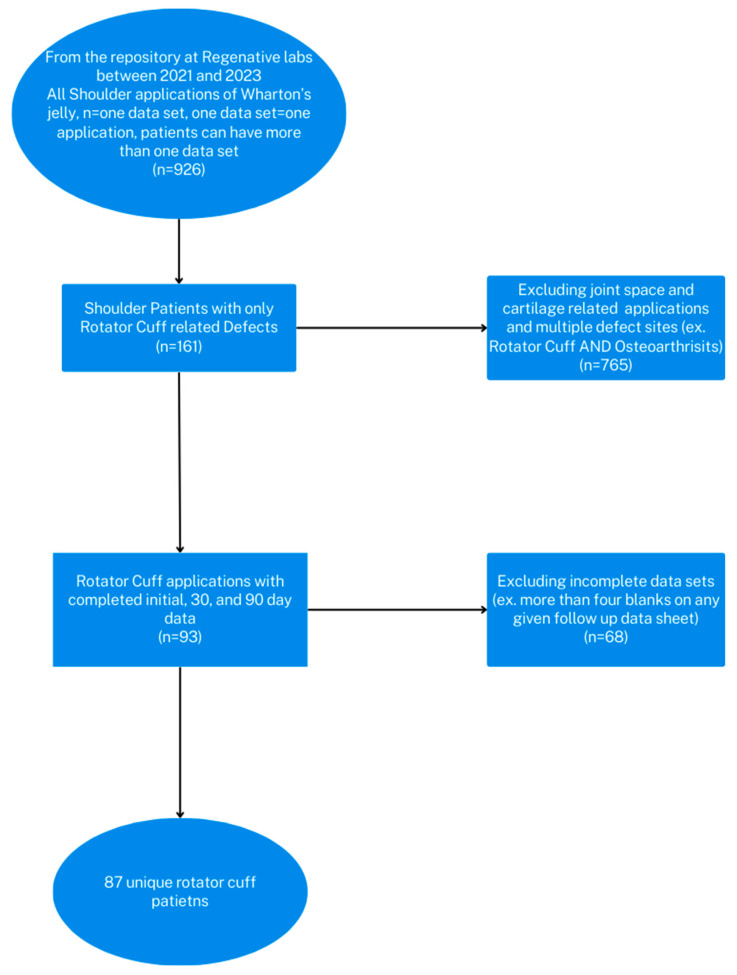
Flowchart of the inclusion and exclusion criteria applied to the repository when selecting a cohort for this study.

**Figure 2 biomedicines-12-00710-f002:**
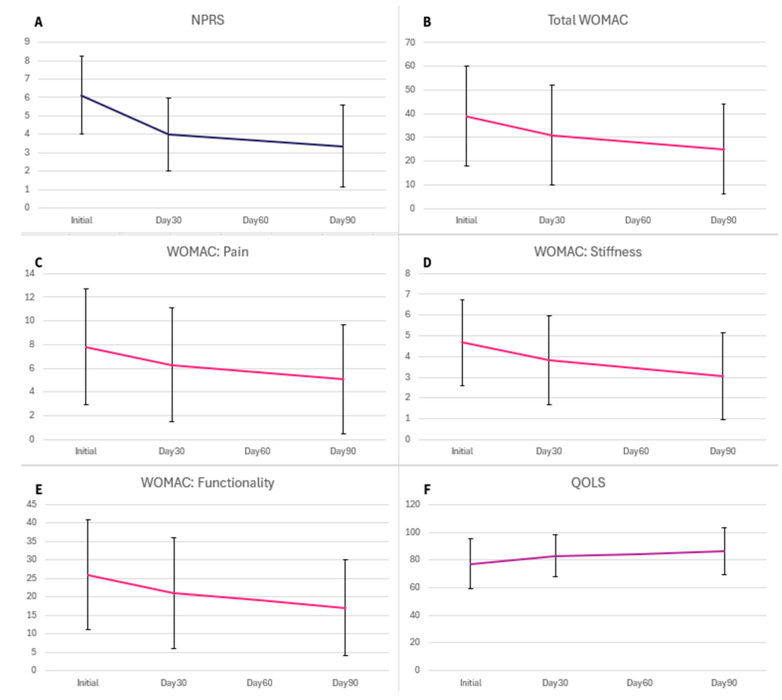
Average score improvement in each scale.

**Figure 3 biomedicines-12-00710-f003:**
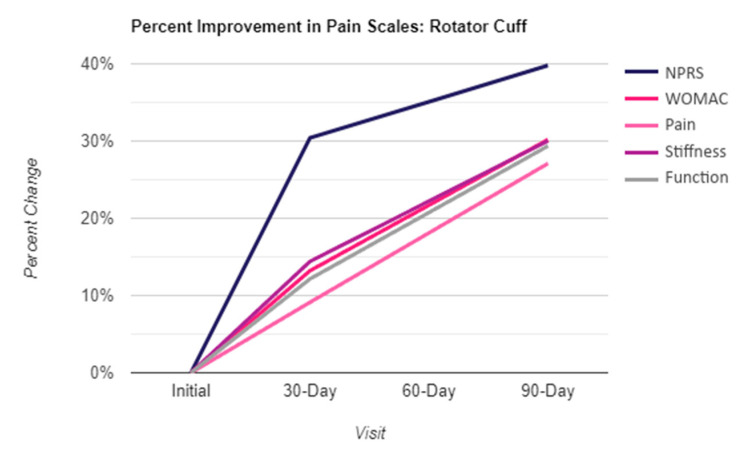
Percent improvement in five scales based on patient-reported scores.

**Figure 4 biomedicines-12-00710-f004:**
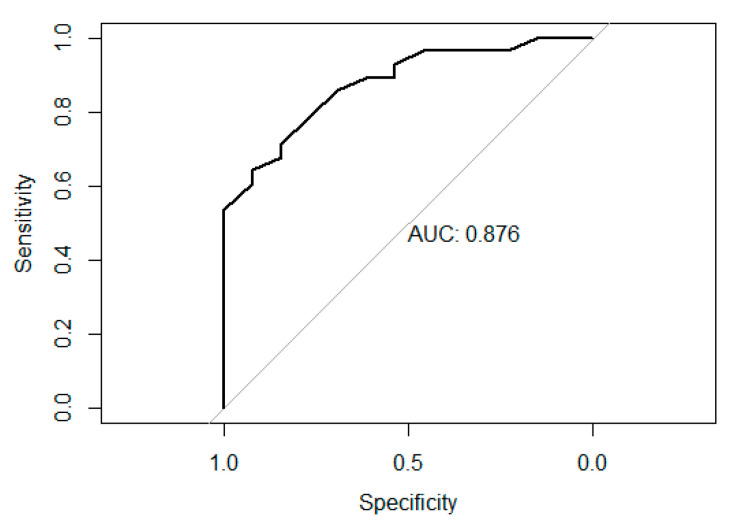
ROC plot for WOMAC.

**Figure 5 biomedicines-12-00710-f005:**
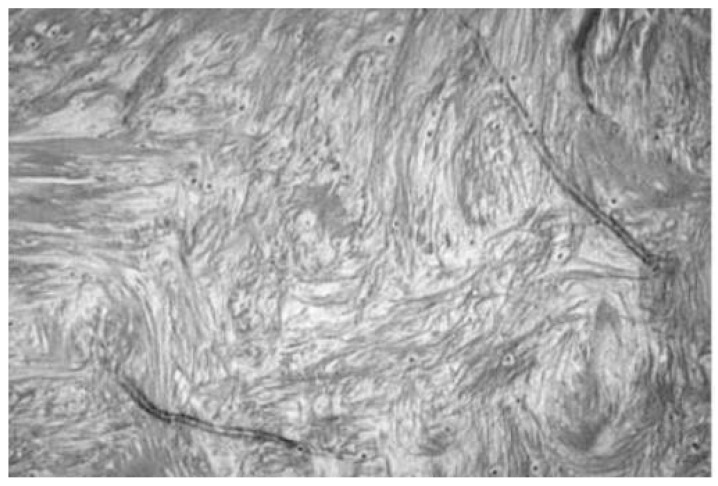
An example of shoulder tendon tears from Hashimoto et al. in 2003. A large tear in a 74-year-old man shows thin and disoriented collagen fibers in the torn tendon (Stain, Masson trichrome; original magnification, ×10) [30].

**Figure 6 biomedicines-12-00710-f006:**
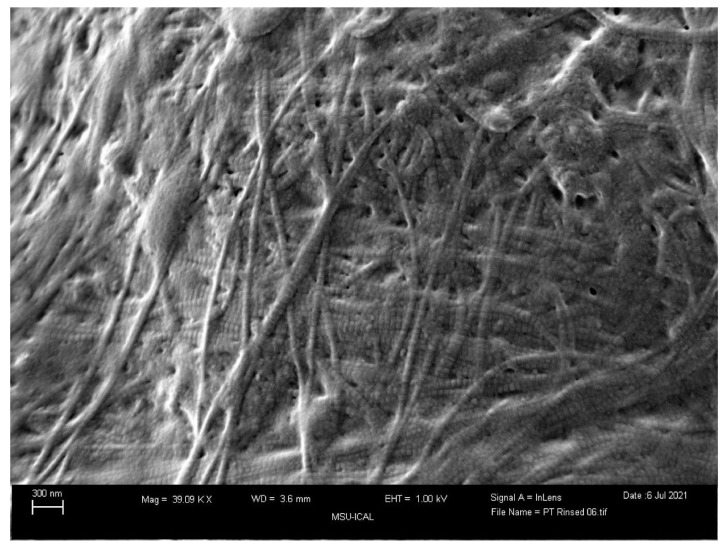
Scanning electron microscopy image of the Wharton’s jelly tissue product used.

**Table 1 biomedicines-12-00710-t001:** Patient characteristics.

Characteristic	N	*n* = 87 ^1^
Age in Years	84	Mean(SD):71 (10) Min: 36 Max: 89
Missing		3
BMI in kg/m^2^	56	Mean(SD):27.7 (4.7) Min: 17.1 Max: 38.6
Missing		31
Gender	87	
Female		42 (48%)
Male		45 (52%)
Missing		0

^1^ Mean (SD) Minimum Maximum; *n* (%).

**Table 2 biomedicines-12-00710-t002:** Sample size and mean (SD) of six scales for each interval.

Interval	N	NPRS ^1^	N	WOMAC ^1^	N	Pain ^1^	N	Stiffness ^1^	N	Functionality ^1^	N	QOLS ^1^
Initial	75	6.13 (2.10)	87	39 (21)	87	7.8 (4.9)	87	4.67 (2.06)	87	26 (15)	80	77 (18)
Day 30	73	3.99 (2.00)	87	31 (21)	87	6.3 (4.8)	87	3.84 (2.14)	87	21 (15)	78	83 (15)
Day 90	67	3.37 (2.20)	87	25 (19)	87	5.1 (4.6)	87	3.06 (2.10)	87	17 (13)	79	86 (17)

^1^ Mean (SD).

**Table 3 biomedicines-12-00710-t003:** Results of Tukey’s test for six pain scales between intervals.

Scales	Interval	Difference	95% CI	*p* Value
NPRS	Day 30–Initial	−2.147	−2.96, −1.33	0.000 *
	Day 90–Initial	−2.76	−3.59, −1.93	0.000 *
	Day 90–Day 30	−0.613	−1.45, 0.23	0.199
WOMAC	Day 30–Initial	−7.414	−14.77, −0.05	0.048
	Day 90–Initial	−13.506	−20.87, −6.15	0.000 *
	Day 90–Day 30	−6.092	−13.45, 1.27	0.127
Pain	Day 30–Initial	−1.494	−3.19, 0.2	0.097
	Day 90–Initial	−2.724	−4.42, −1.03	0.001 *
	Day 90–Day 30	−1.23	−2.93, 0.47	0.205
Stiffness	Day 30–Initial	−0.828	−1.58, −0.08	0.027 *
	Day 90–Initial	−1.655	−2.41, −0.9	0.00 *
	Day 90–Day 30	−0.828	−1.58, −0.08	0.027 *
Functionality	Day 30–Initial	−5.092	−10.31, 0.13	0.058
	Day 90–Initial	−9.126	−14.35, −3.91	0.000 *
	Day 90–Day 30	−4.034	−9.26, 1.19	0.165
QOLS	Day 30–Initial	6.22	−0.08, 12.51	0.054
	Day 90–Initial	8.9	2.62, 15.17	0.003 *
	Day 90–Day 30	2.68	−3.63, 8.99	0.577

* *p*-values less than 0.05.

**Table 4 biomedicines-12-00710-t004:** *p*-values of ANOVA for pain scales between gender, age, and BMI.

Covariates	NPRS	WOMAC	Pain	Stiffness	Functionality	QOLS
Age	0.723	0.195	0.124	0.374	0.217	0.276
Gender	0.380	0.844	0.866	0.430	0.924	0.394
BMI	0.942	0.549	0.731	0.880	0.467	0.849

For *p*-values > 0.05, there is no difference in changes among age, BMI, or gender.

**Table 5 biomedicines-12-00710-t005:** Descriptive statistics for anchor question.

Scales	N	Not Better, N = 13 ^1^	Slightly Better, N = 28 ^1^	Better, N = 14 ^1^	Much Better, N = 11 ^1^
WOMAC	66	−6 (8), −25, 4	10 (16), −14, 68	30 (20), 4, 68	30 (18), 5, 63
Pain	66	−1.8 (3.3), −8.0, 2.0	2.4 (3.7), −3.0, 16.0	6.2 (4.9), 0.0, 16.0	5.8 (4.2), −1.0, 11.0
Stiffness	66	−0.54 (1.33), −3.00, 1.00	1.21 (1.79), −2.00, 6.00	3.64 (2.10), 0.00, 7.00	3.45 (2.11), 0.00, 7.00
Functionality	66	−4 (6), −14, 2	6 (11), −12, 46	20 (14), 1, 45	20 (13), 3, 45
QOLS	59	1 (7), −9, 14	−3 (10), −22, 25	−21 (24), −64, 12	−20 (23), −67, 14

^1^ Mean (SD) Minimum Maximum.

**Table 6 biomedicines-12-00710-t006:** AUC, sensitivity, specificity, and Youden’s index for five scales.

Name	WOMAC	Pain	Stiffness	Functionality	QOLS
AUC	0.88	0.79	0.79	0.83	0.61
Sensitivity	0.64	0.86	0.68	0.57	0.85
Specificity	0.92	0.54	0.77	1.00	0.40
Youden’s Index	0.57	0.40	0.45	0.57	0.25

If the AUC > 0.7, the estimation is useful.

**Table 7 biomedicines-12-00710-t007:** MCID and percentage exceeding the MCID.

Scales	MCID_AUC_	MC_p_total_	% of Exceed MCID_AUC_	% At-Least One Unit Improved
WOMAC	10.00	13.51	51.7	72.4
Pain	2.36	2.72	43.7	66.7
Stiffness	1.21	1.66	41.4	65.5
Functionality	6.43	9.13	50.6	73.6
QOLS	−2.81	−9.12	48.3	57.1

MCID_AUC_: mean changes for “Slightly better” anchor group using ROC curve method. MC_p_total_: mean changes for all patients. At least one unit improved: one or more scores improved higher than Initial.

## Data Availability

Data available upon request.

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
