# Peer review of "Safety and Efficacy of Wharton’s Jelly Connective Tissue Allograft for Rotator Cuff Tears: Findings from a Retrospective Observational Study"

_biomedicines, 2024, doi:10.3390/biomedicines12040710_

Round 1
Reviewer 1 Report
Comments and Suggestions for Authors
In this study the researchers observed the efficacy and safety of using Wharton's jelly tissue allografts applied to structural defects of the rotator cuff in males and females aged around 71 years.
The Introduction provides a clear and comprehensive background for the study.
Line 30: State whether these injury rates are national or international.
The aim is clearly stated. However, I suggest moving it to the end of the paragraph rather than the beginning.
Line 103: It might be better to say that "the population was made up of 42 females and 45 males" Rather than "divided into..."
The Methods and Results sections are very well written.
Move all figure legends to below the figures. Table descriptions should remain above each table.
Keep consistent with figure nomenclature. Figures A and B should be 3 and 4.
Author Response
Dear reviewer, thank you so much for taking the time to review our paper. Your comments were insightful and extremely helpful. We incorporated all of your edits and feel our paper is much stronger. Please see our responses to your comments below.
- We have clarified that this stat is for the United States
- We have moved the aim to the end of the paragraph.
- We edited this line to read as you suggested.
- We corrected the layout of the figure descriptions
- We have corrected the figure nomenclature to be all numbers.
Reviewer 2 Report
Comments and Suggestions for Authors
Thank you for the opportunity to review your manuscript, “A Retrospective Study of Wharton’s Jelly Tissue Allografts for Rotator Cuff Defect Applications.”
The manuscript's subject matter is appropriate; however, the main problem is the absence of a comparison group to contrast the changes. Moreover, as the authors rightly say, only partially appropriate variables for the study have been included.
Line 25-26. Specify that it is an alternative in the control of symptomatology.
Line 72-75. The objective should be at the end of the introduction, as the subsequent explanation should precede the objective.
Material and methods should start with the study design used.
The material and methods begin with a mixture of statistical analysis and ethical considerations. Each should go into its section.
I don't understand that 36-year-old patients are included and that, in line 69, a justification is made about the elderly.
Table 1 should be better explained. The range is presented without separation between values or indication of what it is, leading to confusion.
There is no statistical analysis section.
As there is no control group, we cannot compare whether the changes are due to the time factor, the intervention or another factor. The design could lead to conclusions that magnify the technique.
This study is based on pain but presents scales that are not pain-specific. Subsequently, the authors argue the possible problems with these scales.
Line 238-239. Statistically significant improvement in pain does not imply that it is safe, and even less so if it is not compared to another intervention.
Line 216-226. Statistically significant improvement in pain does not imply that it is safe, and even less so if it is not compared to another intervention.
The discussion is generally poor and does not focus on discussing the results with other studies but instead on reinforcing the authors' ideas about the article, with few references.
The main limitation is that there is no comparison group. It should be included in the limitations.
The conclusion should be lowered, as to say that it has been proven with the current study design is too optimistic.
Line 284. The term "extremely significant" is misleading. It should be rephrased.
Line 289-291. This argument should be passed on for discussion as an opinion of the authors. To draw this conclusion from the study is not appropriate.
Author Response
Dear reviewer, thank you so much for taking the time to review our paper. Your comments were insightful and extremely helpful. We incorporated all of your edits and feel our paper is much stronger. Please see our responses to your comments below.
- Wharton’s Jelly applications aim to supplement and, therefore, repair the structural tissue defects associated with rotator cuff disorders, targeting the root problem of the pain rather than just the control of symptoms.
- We have moved the objective to the end of the paragraph.
- We have restructured the methods and materials section to begin with the study design. This now includes a more specific explanation of where the data came from and how it was sorted with a retrospective study flow chart.
- We have created distinctions between the study design, which includes the ethical approval and the demographics of the patients, and separated them into separate sections.
- We have removed the justification from line 69 regarding elderly patients to keep the introduction consistent in representing a wide range of patients.
- The mean(sd), minimum, and maximum have been labeled in each section to avoid any confusion in the patient demographics.
- We have added a statistical analysis section to the methods and discussion.
- Although we did not have a control group we added comparisons to other studies and meta-analysis of similar patient groups to provide context for our results and how they might compare to other treatments and placebos in future studies.
- By utilizing the NPRS (Numeric Pain Rating Scale), we are able to evaluate a patient’s level of pain as a quantitative value. The use of this scale allows for a patient's pain level to be assessed in a non-biased manner. Some concern was addressed with the utilization of the WOMAC scale, as it is designed for the function of the lower extremities rather than the upper extremities, such as the rotator cuff. This limitation was addressed in lines 315-317. The WOMAC scale is utilized primarily to evaluate the patient’s improvement in overall function.
- We have corrected this statement to include the fact that there were no adverse reactions reported upon application or throughout the study, which implies the safety of Wharton’s jelly.
- Throughout lines 216-226, the statistics for the anchor question were described. NPRS was discussed to describe how the patient’s pain was evaluated in this study. The article did not state that statistically significant improvement in pain equates to the safety of the product. To further address the safety of the product, lines 289-292 discuss how no immune responses were elicited and no adverse reactions were reported, ultimately demonstrating the safety of WJ. In line 331, we further addressed the need for additional research to confirm the safety and efficacy of the use of WJ.
- We have added several paragraphs to the discussion referencing articles that can be used as a retrospective control group and a comparison of other possible treatment methods. Following that paragraph, another paragraph was added to the discussion section to address the national healthcare costs of rotator cuff repairs.
- We have added a discussion of the limitation of no control group.
- We have rephrased the conclusion to sound less definitive and more representative of these observations based on this cohort.
- We have rephrased this sentence in the conclusion to read clearly that these findings are most noteworthy for the elderly population because of their slower recovery times.
- We have made our closing remark more open as we believe our study and other studies regarding WJ for other structural defects do lay the groundwork for using WJ in other defects around the body.
Reviewer 3 Report
Comments and Suggestions for Authors
This manuscript does not hold the utmost priority for publication. My perspectives are articulated below:
1. Design: The study ought to incorporate a control group, and even a retrospective control group would be preferable to none.
2. Lines 121–122: The affected anatomy appears excessively heterogeneous, encompassing the supraspinatus tendon, biceps tendon insertion, labral tear, and subscapularis tear. I would recommend concentrating solely on the supraspinatus tendon.
3. Figure 1: It warrants consideration whether the improvement of the scales should be expressed in percentages. The distinction in percentage terms between the improvement of NPRS from 9 to 8 and from 2 to 1 is noteworthy.
4. Lines 188–189: The demarcation between "not better" or "slightly better" necessitates discussion. For Japanese or Taiwanese patients, "slightly better" may imply "not better," and "much better" might actually signify "slightly better."
5. Lines 216–226: The ANOVA test and Turkey post hoc test are exceedingly commonplace. It is worth contemplating whether the authors need to dedicate a paragraph to elucidate such widely used statistical tests.
Comments on the Quality of English LanguageMinor editing of English language required.
Author Response
Dear reviewer, thank you so much for taking the time to review our paper. Your comments were insightful and extremely helpful. We incorporated all of your edits and feel our paper is much stronger. Please see our responses to your comments below.
- Although we did not have a control group we added comparisons to other studies and meta-analysis of similar patient groups to provide context for our results and how they might compare to other treatments and placebos in future studies.
- The previous line stating that the “supraspinatus tendon, biceps tendon insertion, labral tear, and subscapularis tear” were the most common application sites has been corrected to just “the supraspinatus and subscapularis.” This was a miscommunication error between our team as those are the most common shoulder applications in one physician's practice but do not accurately represent the rotator cuff-specific cohort selected for this observational retrospective study.
- Figure 2: The percentages calculated in Figure 1 are simply based on the average scores of the cohort, representing what level of improvement patients might expect regardless of the starting score. An important note to add is that as a patient outcomes-focused clinic, 40% improvement is not the goal, 100% is. So, this singular application might not be the end of care for someone who goes from a 9 to an 8, but it is for the patient who goes from a 2 to a 1, we have expanded upon this idea in the discussion. Arguably, the most important calculation is if the patients were happy with their care, which we tried to determine with the MCID. The difference between going from a 9 to an 8 and a 2 to a 1 is an important distinction when considering the value of care for a patient. It warrants consideration whether the improvement of the scales should be expressed in percentages. The distinction in percentage terms between the improvement of NPRS from 9 to 8 and from 2 to 1 is noteworthy.
- The demarcation between "not better" or "slightly better" was not a question asked to patients directly but rather rages that we calculated statistically based on patients' reported scores in their NPRS and WOMAC.
- We have edited the paragraphs that were previously solely describing the statistical tests to ones that contain a discussion about our results.
Reviewer 4 Report
Comments and Suggestions for Authors
The authors present the results of a retrospective pilot study on the use of umbilical cord concentrates for infiltrative treatment of rotator cuff lesions in an elderly population.
1) Title: The authors' proposed title lacks information about the study. I suggest a more fitting title i.e.: "Safety and Efficacy of Wharton's Jelly-Derived Mesenchymal Stem Cell Injections for Rotator Cuff Tears: Findings from a Retrospective Non-Controlled Study".
2) Introduction: The study's objectives and the question it aims to answer should be clearly stated in the introduction.
3) Methods: Details regarding patient recruitment, inclusion and exclusion criteria, and specific indications for treatment should be clearly stated. Additionally, baseline characteristics of patients, such as the type and size of the lesion, degree of arthritis, and associated injuries, should be detailed.
4) In the methodology section, provide details on how Wharton's jelly is collected, processed, and prepared for use, and how it is applied to the rotator cuff lesion (e.g., ultrasound-guided infiltration).
5) Clearly define the abbreviations NPRS and QOLS in the methodology section.
6) The decontextualized sentence at lines 95-96 should be placed in a subsection on statistical analysis. Furthermore, considerations regarding patient satisfaction, MCID and AUC analyses in the results should be explained in the methods and mentioned in the abstract and title.
7) The discussion should focus on the study's objectives, including consideration of non-surgical treatments for patients unresponsive to conventional treatments and other treatments of regenerative medicine.
a. Mocini F, et al. The role of adipose derived stem cells in the treatment of rotator cuff tears: from basic science to clinical application. Orthop Rev (Pavia). 2020;12(Suppl 1):8682. doi:10.4081/or.2020.8682.
b. Bhan K, Singh B. Efficacy of Platelet-Rich Plasma Injection in the Management of Rotator Cuff Tendinopathy: A Review of the Current Literature. Cureus. 2022;14(6):e26103. doi:10.7759/cureus.26103.
8) Finally, discuss the study limitations, including the lack of consideration of the placebo effect and the methodologies used to mitigate it. Additional considerations on cost-effectiveness and product availability should be included. If the goal is to determine the MCID of various patient-reported outcomes, cite validation studies of these instruments in patients with rotator cuff lesions.
Comments on the Quality of English Languageminor editing is required.
Author Response
Dear reviewer, thank you so much for taking the time to review our paper. Your comments were insightful and extremely helpful. We incorporated all of your edits and feel our paper is much stronger. Please see our responses to your comments below.
- We have changed the title to a more descriptive title “Safety and Efficacy of Wharton's Jelly Connective Tissue Allo-graft for Rotator Cuff Tears: Findings from a Retrospective Observational Non-Controlled Study” The Wharton’s jelly was not processed to extract or culture cells, the entire connective tissue was cut and suspended in saline for direct tissue supplementation, therefore is classified as a tissue allograft, not a stem cell injection.
2. The aim of the study is now clearly stated in the last sentence of the last paragraph of the introduction section.
3. We have further clarified the methods used in this retrospective study, including the details about the repository data form and the inclusion and exclusion criteria flow chart for the study. The types of lesions included and excluded have also been listed.
4. The method section now includes more specific details of how the WJ tissue allograft arrives at the clinic. It is purchased from Regenative Labs, so the manufacturing details are available directly from them. However, we provided as many specifics as possible about the vials of tissue and what is in them, and that the lab is certified by the FDA and AATB. We also clarified that the WJ was applied to the affected area under ultrasound guidance.
5. We have now clearly defined the abbreviations NPRS and QOLS in the methodology section.
6. We edited the methods section to include statistical analysis and reorganized the whole section to be more clear. We also added a comment about using MCID for patient satisfaction calculations in the abstract.
7. We pulled several additional studies to cover some other modalities and how they compare to each other.
8. Although we did not have a control group we added comparisons to other studies and meta-analysis of similar patient groups to provide context for our results and how they might compare to other treatments and placebos in future studies. We also added a cost analysis section and cited our validation sources for the statistical analysis portion.
Round 2
Reviewer 2 Report
Comments and Suggestions for Authors
The authors have answered all my questions and addressed the necessary changes.
However, two minor issues should be reworded.
Line 171 - 1174- It is unnecessary to explain what ANOVA is used for and why a post-hoc test is used. The test used above is already mentioned.
Line 175 - 179- It is not appropriate to state what questions the statistical analysis is intended to answer, but mentioning what type of analysis was performed is sufficient. The questions to be answered are the study's objectives or specific objectives.
Author Response
Thank you for accepting our previous edits. We have corrected the two minor comments you made in this round.
1. We have simplified this section so as not to overexplain the tests used.
2. We have removed the questions from the statistical analysis portion of the methods section.
Reviewer 3 Report
Comments and Suggestions for Authors
Please see the attached file.

Author Response
Thank you greatly for your approval of our previous edits. We have worked to address all your additional comments below.
- Although not from our cohort, we feel the added visual representation is helpful to compare with the WJ tissue allograft. However, we have added additional clarification in the figure's description that it is from a different study.
- We have double-checked each table to ensure no words are split amongst lines.
- We have added graphs for each scale individually from the actual score averages.
- We have removed the quotation marks.
- We now include more specific examples of the current conservative treatments available in the introduction.
- When this study was initially designed, the primary use of WJ was for cartilage supplementation in the knees, so while it was left open-ended to accept data on any homologous use site, the generic data form for knees was used by most clinics. As we renew the study each year, we have begun implementing data scoring sheets specific to each use site but wanted to publish this paper on our initial data to hopefully promote more frequent use in the shoulder so we can then expand our study and have a more refined and robust group to observe as well as finding control groups.
Reviewer 4 Report
Comments and Suggestions for Authors
the article has overall improved following the revisions made, and it is now potentially suitable for publication.
Author Response
We thank you greatly for your time and effort in assisting us in improving this manuscript.
Round 3
Reviewer 3 Report
Comments and Suggestions for Authors
The revised manuscript demonstrates significant improvement. I offer a few suggestions for your consideration:
1. I can appreciate the rationale for not incorporating shoulder-specific functional scores. It would be beneficial to explicitly acknowledge this absence in the limitations section.
2. In Section 2.5 and Table 3, it is advisable to clarify the distinction between between-group ANOVA and within-group (difference time points) repeated measure ANOVA. Repeated measure ANOVA appears more fitting for the data presented in Table 3, and this choice could be explicitly articulated.
Author Response
Thank you greatly for your feedback. We have made the additional edits based on your comments.
- We have added a more specific rationale to the limitation discussion section to explain why WOMAC was used instead of more suitable options that we are incorporating into the future of the study.
- The ANOVA used is Repeated Measures ANOVA, which is a within-subject/group ANOVA. We revised the language in Section 2.5 to address the type of ANOVA and added an asterisk for p-values less than 0.05 in Table 3.